# Comparison of Trans-Arterial Chemoembolization and Bland Embolization for the Treatment of Hepatocellular Carcinoma: A Propensity Score Analysis

**DOI:** 10.3390/cancers13040812

**Published:** 2021-02-15

**Authors:** Gaël S. Roth, Maxime Benhamou, Yann Teyssier, Arnaud Seigneurin, Mélodie Abousalihac, Christian Sengel, Olivier Seror, Julien Ghelfi, Nathalie Ganne-Carrié, Lorraine Blaise, Olivier Sutter, Thomas Decaens, Jean-Charles Nault

**Affiliations:** 1Faculty of Medicine, Grenoble-Alpes University, 38043 Grenoble, France; yteyssier@chu-grenoble.fr (Y.T.); ASeigneurin@chu-grenoble.fr (A.S.); evelyne.abousalihac@ch-metropole-savoie.fr (M.A.); jghelfi@chu-grenoble.fr (J.G.); tdecaens@chu-grenoble.fr (T.D.); 2Department of Hepato-Gastroenterology and Digestive Oncology, CHU Grenoble-Alpes, 38043 Grenoble, France; 3Institute for Advanced Biosciences, INSERM U1209, CNRS UMR 5309, Grenoble-Alpes University, 38043 Grenoble, France; 4Functional Interventional Radiology Unit, Avicenne Hospital, Paris-Seine-Saint-Denis University Hospitals, Assistance-Publique Hôpitaux de Paris, 93000 Bobigny, France; maxime.ben-hamou@etu.u-paris.fr (M.B.); olivier.seror@aphp.fr (O.S.); olivier.sutter@aphp.fr (O.S.); 5Imaging Center, Clinical University of Radiology and Medical Imaging, CHU Grenoble-Alpes, 38043 Grenoble, France; csengel@chu-grenoble.fr; 6Medical Assessment Service, CHU Grenoble-Alpes, 38043 Grenoble, France; 7Health Medicine and Human Biology Training and Research Unit, Paris Nord University, 93000 Bobigny, France; nathalie.ganne@aphp.fr; 8Functional Genomics of Solid Tumors Laboratory, Cordeliers Research Center-INSERM UMR 1138, Paris University, 75006 Paris, France; 9Department of Hepatology, Avicenne Hospital, Paris-Seine-Saint-Denis University Hospitals, Assistance-Publique Paris Hospitals, 93000 Bobigny, France; lorraine.blaise@aphp.fr

**Keywords:** hepatocellular carcinoma, trans-arterial chemoembolization, bland embolization, intermediate stage, tumor response, survival

## Abstract

**Simple Summary:**

In this study, efficacy and safety of embolization alone and trans-arterial chemoembolization were compared in 265 patients with intermediate stage hepatocellular carcinoma. Trans-arterial chemoembolization was associated with a significant increase of complete radiological response, but without significant impact on overall response, and survival outcomes after propensity score matching. Both techniques showed similar safety profiles. To this day, embolization alone and trans-arterial chemoembolization are two available options in the treatment of intermediate stage hepatocellular carcinoma.

**Abstract:**

No definitive conclusion could be reached about the role of chemotherapy in adjunction of embolization in the treatment of hepatocellular carcinoma (HCC). We aim to compare radiological response, toxicity and long-term outcomes of patients with hepatocellular carcinoma (HCC) treated by trans-arterial bland embolization (TAE) versus trans-arterial chemoembolization (TACE). We retrospectively included 265 patients with HCC treated by a first session of TACE or TAE in two centers. Clinical and biological features were recorded before the treatment and radiological response was assessed after the first treatment using modified Response Evaluation Criteria in Solid Tumors (mRECIST) criteria. Correlation between the treatment and overall, progression-free and transplantation-free survival was performed after adjustment using a propensity score matching: 86 patients were treated by bland embolization and 179 patients by TACE, including 44 patients with drug-eluting beads and 135 with lipiodol TACE, 89.8% of patients were male with a median age of 65 years old. Cirrhosis was present in 90.9% of patients with a Child Pugh score A in 84% of cases. After adjustment, no difference in the rate of AE, including liver failure, was observed between the two treatments. TACE was associated with a significant increase in complete radiological response (odds ratio (OR) = 8.5 (95% confidence interval (CI): 2.8–25.4)) but not in the overall response rate (OR = 2.2 (95% CI = 0.8–5.8)). No difference in terms of overall survival (*p* = 0.3905), progression-free survival (*p* = 0.4478) and transplantation-free survival (*p* = 0.9020) was observed between TACE and TAE. TACE was associated with a higher rate of complete radiological response but without any impact on overall radiological response, progression-free survival and overall survival compared to TAE.

## 1. Introduction

Liver cancer is the second cause of cancer-related deaths worldwide and is mostly represented by hepatocellular carcinoma (HCC) [1]. Approximately 20% of patients present an intermediate HCC classified as Barcelona Clinic of Liver Cancer (BCLC) stage B for which trans-arterial embolization techniques are the best treatments [2]. As arterial neoangiogenesis is one of the hallmarks of hepatocellular carcinoma (HCC) [3], these procedures lead to tumor ischemia and inhibition of tumor growth through tumor blood flow shut down [4], allowing interesting results in terms of tumor response, around 50% [5]. Nowadays, trans-arterial chemoembolization (TACE), which combines chemotherapy administration followed by embolization, is the standard treatment of BCLC stage B HCC, with a survival improvement compared to best supportive care, based upon a meta-analysis of 5 randomized clinical trials (RCT) with a grade B level of evidence in international guidelines [2,6,7]. Besides, TACE is also sometimes proposed to treat HCC with segmental tumor portal thrombosis, even if its efficacy in this situation remains debated [8], and it is often used in BCLC stage A patients on the waiting list for liver transplantation [9,10].

Nevertheless, the proper action of chemotherapy is poorly described and its interest is still debated as ischemic action of embolization seems to constitute the major part of cytotoxic action [11]. Thus, the superiority of TACE versus trans-arterial embolization (TAE) alone, also called bland embolization, is still controversial. The only RCT comparing TACE versus TAE versus placebo showed an advantage of TACE versus placebo, but the trial was halted prematurely and did not have enough power to conclude about the effect of TAE versus placebo [12].

Other RCT have compared TACE versus TAE and meta-analyses reported that the overall survival and the therapeutic response were similar between the two intra-arterial treatments [13,14,15]. A difference in terms of toxicity between the two regimens of trans-arterial treatments with more adverse events with TACE was also identified in some of these studies [14,16]. Overall, several biases present in these trials decrease the robustness of the results and currently, no definitive conclusion can be reached about the role of chemotherapy in adjunction of embolization in the treatment of non-resectable HCC [13,16].

We aim to compare radiological response, treatment-related toxicity and long-term outcomes in a retrospective bicentric study of patients with hepatocellular carcinoma treated by TAE versus TACE.

## 2. Materials and Methods

### 2.1. Patients Selection

Inclusion criteria were patients presenting HCC diagnosed at histology or using non-invasive criteria at imaging based on European Association for the Study of the Liver (EASL) guidelines, considered not resectable or not amenable to percutaneous ablation by a multidisciplinary tumor board. Patients were Child Pugh score A or B, with a performance status 0 or 1, and no prior trans-arterial procedure. TACE or TAE as a bridge to transplantation was not an exclusion criterion. Exclusion criteria were trans-arterial procedure performed in the treatment of an acute bleeding of HCC and the absence of available pre- and post-procedure imaging to assess the radiological response.

We retrospectively included all patients meeting these criteria in two centers in France: Jean Verdier University Hospital center in Bondy and Grenoble-Alpes University Hospital center: 112 patients treated at Jean Verdier University Hospital from 1 December 2007 until 1 November 2013 were included and 153 patients from the Grenoble University Hospital from 1 June 2011 until 1 December 2014, for a total of 265 patients.

TACE using either drug-eluting beads, doxorubicin or idarubicin [17] was the standard of care for trans-arterial treatment in HCC patients in Grenoble Hospital, whereas TAE using Lipiodol associated with Curaspon was the standard of care for trans-arterial treatment in HCC patients in Jean Verdier Hospital.

The only exception to the systematic use of TAE for HCC treatment in Jean Verdier Hospital was a period of several months when all patients (*n* = 26) received TACE before switching to TAE as the only trans-arterial technic used in the center.

### 2.2. Ethical Approval

Every patient gave their written consent before trans-arterial procedures and study ethics was approved by an independent Institutional Review Board in France (number CRM-2004-084, approval date 25 May 2020).

### 2.3. Trans-Arterial Procedures

Patients were treated with trans-arterial therapy following standard local protocol. Each indication of TACE or TAE was validated during multidisciplinary tumor board including at least an hepatologist, an interventional radiologist and a liver surgeon. First, diagnostic arteriography was performed under local anesthesia, through the right femoral artery, followed by trans-arterial therapy, as selective as possible according to tumor localization and number.

In case of TACE, 50 mg of injectable lyophilized Doxorubicin (Adriblastina^®^, Pfizer Pharma, New York, NY, USA) or 10 mg of injectable lyophilized Idarubicin (Zavedos^®^, Pfizer Pharma, New York, NY, USA) were either manually emulsified with 5–10 mL of iodized oil (Lipiodol^®^ Ultra Fluide, Guerbet, France) or loaded on 100 µm drug-eluting beads (100 μm; Embozene Tandem^®^ microspheres, Celonova Biosciences, Ulm, Germany), as previously described [17]. In Lipiodol TACE, drug administration was immediately followed by embolization using an absorbable gelatin sponge (Curaspon^®^, Curamedical, Assendelft, The Netherlands) to obtain an arterial flow stop during 10 min, under fluoroscopic control.

In case of TAE, 10–15 mL of pure iodized oil (Lipiodol^®^ Ultra Fluide, Guerbet), without emulsion with chemotherapy, was injected through the catheter as selective as possible. This “lipiodolization” was followed by embolization using an absorbable gelatin sponge (Curaspon^®^, Curamedical) until complete stasis of the arterial flow.

As recommended by European guidelines [2], for patients with progressive disease or degradation of liver function, no additional treatment by TAE/TACE was performed. For all other patients, additional TAE/TACE treatment was performed on demand, after discussion at a weekly multidisciplinary tumor board.

### 2.4. Data Collection

All imaging examinations were archived in a picture archiving and communication system (PACS, Agfa HealthCare^®^, Mortsel, Belgium). Medical parameters as well as biological data were extracted from the patients’ electronic medical records and independently reviewed. Radiological reviewing was realized for this study blindly to clinical data by two radiologists (Olivier Sutter and Yann Teyssier).

The following patient characteristics were collected: sex, age, World Health Organization (WHO) performance status, body mass index (BMI), etiology of chronic liver disease and cirrhosis status. Cirrhosis was diagnosed by biopsy or using non-invasive methods (transient elastography or blood tests). Several biological variables were recorded at inclusion: albumin, prothrombin time, bilirubin, transaminases, gamma glutamyl transferase, alkaline phosphatase, creatinine, platelets and alpha-fetoprotein (AFP). Every patient had a pre-operative imaging, and a post-operative imaging, within 3 months after TACE as recommended [2], to assess tumor response and determine the need to perform an additional TACE or not. Radiological examination was performed with multiphasic liver magnetic resonance imaging (MRI) or computed tomography (CT) scan at baseline and after 6 to 8 weeks after the first session of treatment. Imaging data collected were tumor number, size of the largest nodules and presence of portal vein invasion on preoperative imaging. Tumor response was assessed according to the modified Response Evaluation Criteria in Solid Tumors (mRECIST) criteria with complete response (CR), partial response (PR), stable disease (SD) and progressive disease (PD) [18]. Objective response rate (ORR) corresponded to complete and partial radiological response.

During the post-embolization period, adverse events (AEs), throughout two months following the treatment, were recorded based on clinical examinations, systematic biological follow-up (renal and hepatic functions) and imaging follow-up. We graded the AE from grade 1 to 5 according to the Common Terminology Criteria for Adverse Events (CTCAE) v5.0. After the first treatment, all patients were prospectively followed-up until death or the last recorded visit, until 30 June 2018.

### 2.5. Statistical Analysis

Categorical variables such as tumor response rate and adverse events were compared using exact Chi-square tests. Survival outcomes such as progression-free survival (PFS), overall survival (OS) and liver transplant-free survival (LTFS) were computed using the Kaplan–Meier method, and Log-rank tests were used to compare survival rates. Overall survival was calculated from the date of first treatment to the date of death or last recorded visit and data were censored at the date of liver transplantation. Progression-free survival was calculated from the date of first treatment to the date of death, date of radiological progression or last recorded visit. Transplantation-free survival was calculated from the date of first treatment to the date of death, date of liver transplantation or last recorded visit.

Logistic regressions were used to compare binary variables. Mixed linear models were used to assess differences in the evolution of the Child Pugh score TAE and TACE. For each analysis, comparisons were performed with and without weighting by a propensity score (inverse probability of treatment weighting (IPTW)) to adjust for confounding factors. Statistical significance is expressed with *p*-value for univariate analyses and odds ratio (OR) with 95% confidence interval (95% CI) for propensity score weighted results, including the age, Child Pugh score, AFP level, the sum of the 2 main liver nodules and the number of tumors. Analyses were performed using Stata version 16.1 (Stata Corporation, College Station, TX, USA).

## 3. Results

### 3.1. Clinical Features of the Population

A total of 265 patients were included with a median follow-up of 21.7 months: 86 patients were treated by TAE and 179 patients by TACE, including 44 patients with DC beads and 135 with Lipiodol TACE. Chemotherapy used in TACE was doxorubicine and idarubicine in respectively 110 and 25 patients. The median number of sessions of TACE and TAE during follow-up was 2 (IQR (interquartile range) = 1–2) in each group. All patients treated by TAE were treated in Jean Verdier hospital and 86% of patients treated by TACE were treated in Grenoble Hospital.

Two-hundred and thirty-eight patients (89.8%) were male with a median age of 65 years old. Cirrhosis was present in 241 patients (90.9%) with Child Pugh A in 83.8% of cases. Etiologies of underlying chronic liver diseases are detailed in Table 1. Twenty-eight percent of patients were classified BCLC-A, 57% BCLC-B and 26% BCLC-C, with mostly segmental portal tumor invasion. Patients treated by TAE were older (median age: 69 versus 63; *p* = 0.0003), had a higher tumor burden (median tumor number: 3 versus 2; *p* < 0.0001), with less BCLC A HCCs (10.5% versus 35.8%; *p* < 0.0001), compared to patients treated by TACE. In contrast, patients treated by TAE had a lower Child Pugh score (median: 5 versus 6; *p* < 0.0001) and a lower AFP level (median: 7.0 versus 15.0 ng/mL; *p* = 0.0045) compared to patients treated by TACE (Table 1). Fifty-one patients (19.3%) had liver transplantation after a median follow-up of 21.6 months, with more patients treated by TACE receiving transplantation (24.6%) than patients treated by TAE (8.1%, *p* = 0.001). Other treatments following TAE and TACE are detailed in Table 1, as well as the different characteristics of the population.

### 3.2. Adverse Events Related to Trans-Arterial Treatment

Adverse events occurring within 2 months following the trans-arterial treatment were observed in 25.7% of patients, including mainly fatigue, pain, biliary complications and liver failure. Without any adjustment, all grade AE were more frequent in patients treated by TACE compared to patients treated by TAE (29.6% and 17.4% respectively, OR = 2.0 (95% CI = 1.01–3.8)), whereas the incidence of grade 3 and 4 AE was not significantly different. After weighting by propensity score (including the age, Child Pugh score, AFP level, the sum of the 2 main liver nodules and the number of tumors), TACE was no more significantly associated with a higher rate of all grade AE (OR = 2.3 (95% CI: 0.8–6.3).

Child Pugh score variation at the first radiological assessment did not differ significantly between groups with an increase of +0.6 (95% CI: +0.3; +0.9) after TAE and +0.4 (95% CI: +0.2; +0.6) after TACE (*p* = 0.2095). Similar results were obtained after weighting by propensity score with an increase of the Child Pugh score of +0.8 (95% CI: +0.2; +1.3) after TAE and +0.4 (95% CI: +0.3; +0.6) after TACE (*p* = 0.2267). When restricting the analysis to cirrhotic patients, no difference of the increase of the Child Pugh score was observed without (*p* = 0.3345) and with propensity score weighting (*p* = 0.5032).

### 3.3. Radiological Response

Based on mRECIST criteria, ORR after the first session of TACE/TAE was 63.3% (including 23.5% of complete and 39.8% of partial response), 21.2% of patients had stable disease and 15.5% a progressive disease at imaging. In the whole population, a significant increase in OS was observed in patients presenting a tumor response (CR and PR) with a median survival time of 32.1 months versus 20.1 months in non-responders (*p* = 0.0049). Also, higher PFS (*p* = 0.00001) and LTFS (*p* = 0.0009) were observed in responders (CR and PR) versus non responders (SD and PD) (Figure 1).

Without any adjustment, TACE was associated with a higher ORR (67.4%) compared to TAE (54.6%, Chi2 test *p* = 0.044). After propensity score weighting, TACE was no more significantly associated with a higher rate of radiological response ORR (OR = 2.2 (95% CI = 0.8–5.8)) but remains significantly associated with a higher rate of complete response (OR = 8.5 (95% CI: 2.8–25.4)).

### 3.4. Progression-Free Survival

The median PFS of the whole cohort was 9.3 months. Without weighting, no significant difference between TACE and TAE was observed (median PFS of 9.0 months and 10.8 months respectively, *p* = 0.5010). After weighting by a propensity score, PFS was still not statistically different between patients treated by TACE compared to TAE (*p* = 0.4478) (Figure 2). No significant difference was observed after weighting by a propensity score on PFS in the subgroup of Child Pugh A patients (TACE median PFS = 8.6 months versus TAE 11.7 months, *p* = 0.6201).

### 3.5. Overall Survival

The median overall survival of the whole cohort was 27.7 months, with 76%, 54% and 39% of survival at 1, 2 and 3 years. Without weighting, median overall survival was longer in patients treated by TACE (32.7 months) compared to patients treated by TAE (21.5 months, *p*-value = 0.0009). In contrast, after weighting by a propensity score, no significant difference in terms of overall survival was identified in TACE versus TAE (median: 33.3 versus 28.2 months, respectively; *p* = 0.3508) (Figure 3). Among Child Pugh A patients, a significant increase in survival was observed for patients treated by TACE without weighting (median OS: 33.3 versus 21.0 months; *p* = 0.0001) and after weighting by propensity scores (median OS: 35.0 versus 22.3 months; *p* = 0.0465).

### 3.6. Transplantation-Free Survival

The median transplantation-free survival of the whole cohort was 19.1 months, with 72%, 37% and 22% of survival without transplantation at 1, 2 and 3 years. The median transplantation-free survival was not different in patients treated by TACE (18.7 months) compared to patients treated by TAE (20.1 months, *p* = 0.1512). After weighting by a propensity score, no significant difference in terms of transplantation-free survival was identified in TACE versus TAE (median: 21.0 versus 21.5 months, respectively; *p* = 0.9105) (Figure 4). The same results were observed in Child Pugh A patients without any significant difference after propensity score weighting (*p* = 0.6607).

## 4. Discussion

This study aimed to compare the efficacy and tolerance of TAE compared to TACE in patients with unresectable and non-ablatable HCC based on a retrospective analysis of two tertiary centers in France. Overall, there is limited selection bias between TACE and TAE in each center because TACE was systematically performed in patients from Grenoble and TAE was systematically performed in patients from Jean Verdier Hospital. The only exception was a period of 18 months in Jean Verdier Hospital when all patients received TACE before switching to TAE as the only trans-arterial treatment.

In terms of radiological response using mRECIST criteria, TACE was more efficient to achieve a complete radiological response even after adjustment using a propensity score. These data suggest that the adjunction of chemotherapy with embolotherapy could increase the rate of radiological response. All pre- and post-treatment imagings were reviewed by two independent radiologists in order to better characterize the tumor burden and the objective tumor response. However, one of the limits of our study is the absence of assessment of interobserver agreement about radiological response by two independent reviewers. If the assessment of radiological response (partial versus complete response) after trans-arterial treatment could be sensitive to interobserver variation, progressive disease is considered as more reproductible [19]. In our study, the rate of progressive disease was not significantly different between the TACE group compared to the TAE group.

In terms of impact of radiological response on overall survival, radiological response was associated with longer overall survival. TACE was associated with a higher rate of complete response but without increasing overall survival compared to TAE. This absence of benefit on survival outcomes suggests that complete radiological response is not the only determinant of long-term survival and other parameters such as treatment toxicity, ability to achieve partial radiological response or stable disease, ability to repeat trans-arterial treatment or initiate subsequent treatment by systemic therapy should be taken into account. Moreover, the absence of difference in terms of progressive disease between the two treatments may explain the absence of difference in overall survival.

Besides, TACE was associated with a higher rate of toxicity compared to TAE but the rate of grade 3 or 4 AE was not different. Previous data on toxicity and efficacy have also suggested that TACE was associated with a higher radiological response rate with an increase rate of toxicity [14,16]. This higher rate of toxicity observed in some patients can counterbalance the benefit of radiological response in others as toxicity could impact survival, especially in cirrhotic patients. However, the ability to fully assess the frequency and severity of AE is limited by the retrospective design of our study.

The crude difference observed between the two treatments in terms of raw overall survival may be explained by the higher rate of patients receiving liver transplantation in the TACE group compared to TAE as transplantation-free survival was not different between the two treatments. It is difficult to differentiate the effect of liver transplantation per se with the prognostic impact of the clinical and tumor features of patients amenable to transplantation that have lower tumor burden, a younger age and less comorbidity. The absence of difference in terms of overall survival after propensity score adjustment suggests that patients’ features play a key role in the initial difference between TACE and TAE. However, we cannot exclude that the absence of difference in terms of OS and LTFS between the two groups after adjustment is due to the small size of each group.

These data are important to consider in the therapeutic algorithm of HCC patients which needs to include parameters such as liver function, possibility of liver transplantation and potential treatments available in second line. As our study showed a higher rate of complete response with TACE, and previous published data suggest a higher rate of adverse events and liver deterioration with this technique [16], TACE could be a good option in transplantable patients to optimize tumor control despite an increased risk of toxicity, and TACE could also be a better option for patients potentially accessible to per-cutaneous ablation after downsizing. On the other hand, TAE could be more suitable in a palliative setting, where systemic therapies will be required after treatment failure. These results suggest that we should balance between anti-tumor efficacy and toxicity in HCC patients in order to achieve the best overall survival.

## 5. Conclusions

In conclusion, TACE was associated with a higher complete radiological response rather than TAE but without a significant impact on progression-free and overall survival after adjustment, and with a possible lack of statistic power. Future studies are needed to properly evaluate the effects of the adjunction of chemotherapy to trans-arterial embolization on survival, in patients with HCC.

## Figures and Tables

**Figure 1 cancers-13-00812-f001:**
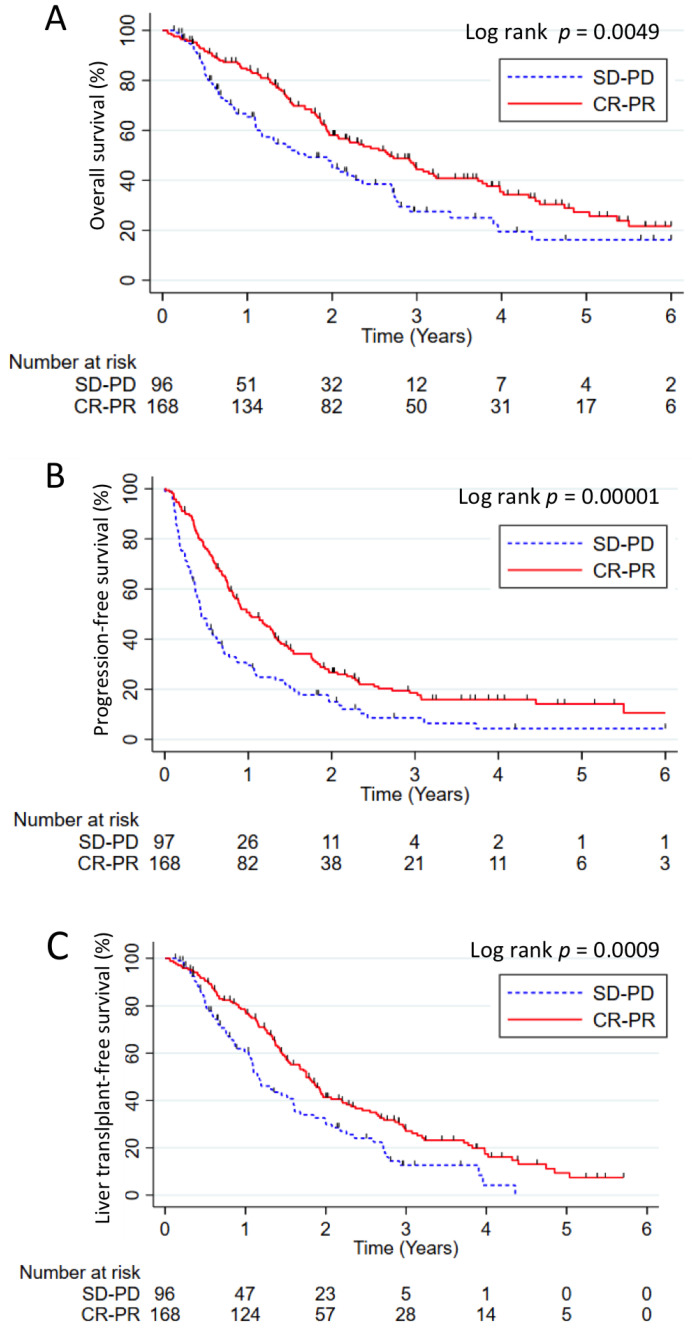
Outcomes according to radiological response. Overall survival (**A**), progression-free survival (**B**) and liver transplantation-free survival (**C**) in patients with radiological response (complete and partial response) versus patients with stable disease or progressive disease according to modified Response Evaluation Criteria in Solid Tumors (mRECIST) criteria. The results were analyzed using the Kaplan–Meier method and compared using the log rank test. Numbers at risk were reported under the *x*-axis.

**Figure 2 cancers-13-00812-f002:**
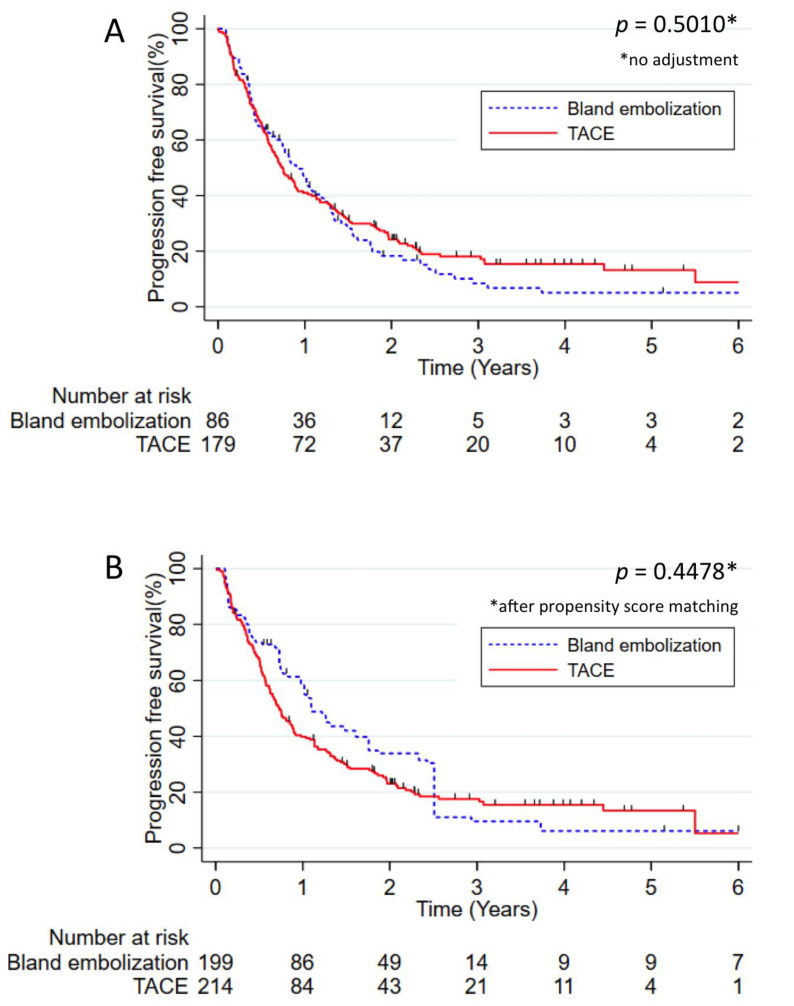
Progression-free survival in patients treated by TACE versus patients treated by TAE. (**A**) Progression-free survival without adjustment and (**B**) progression-free survival adjusted using the propensity score. The results were analyzed using the Kaplan–Meier method and compared using the log rank test. Numbers at risk were reported under the *x*-axis.

**Figure 3 cancers-13-00812-f003:**
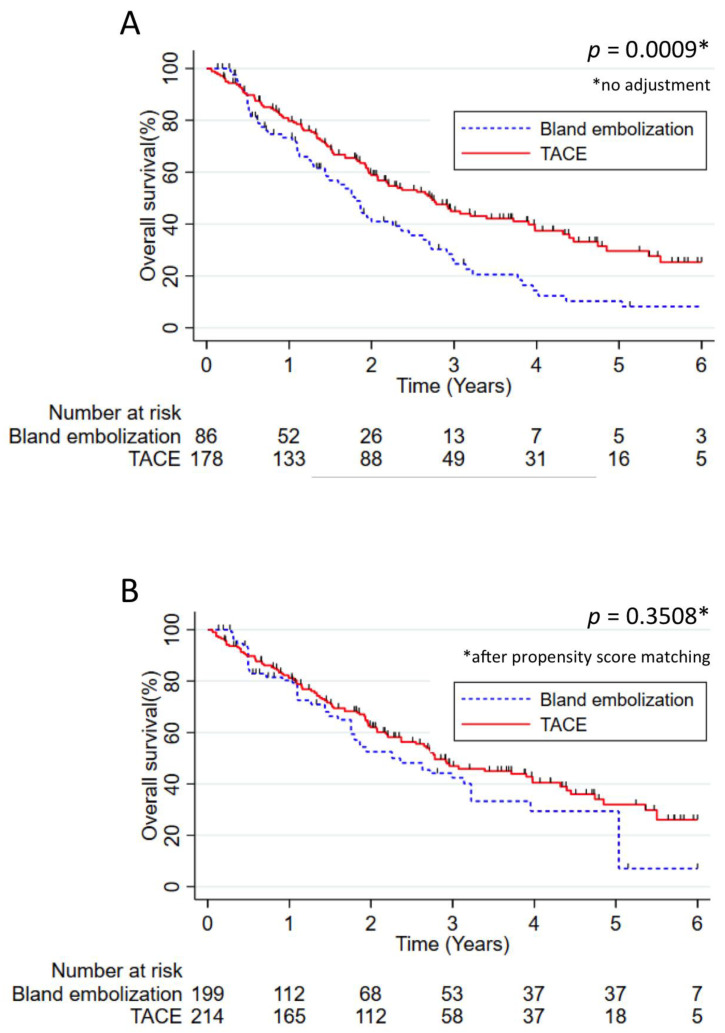
Overall survival in patients treated by TACE versus patients treated by TAE. (**A**) Overall survival without adjustment and (**B**) overall survival adjusted using the propensity score. The results were analyzed using the Kaplan–Meier method and compared using the log rank test. Numbers at risk were reported under the *x*-axis.

**Figure 4 cancers-13-00812-f004:**
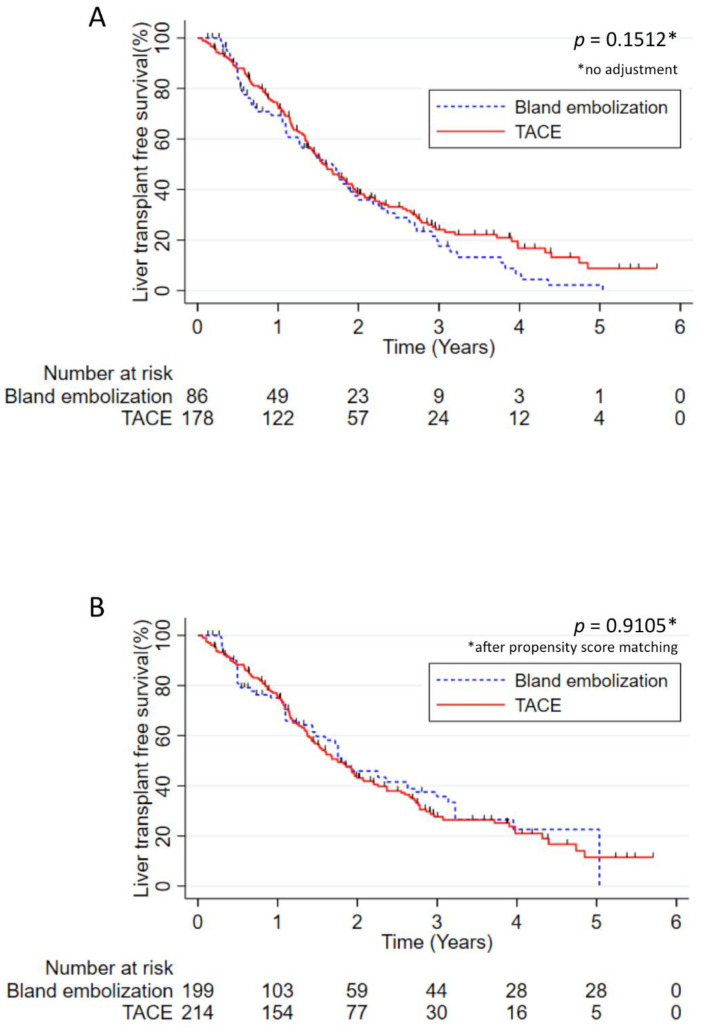
Liver transplantation-free survival in patients treated by TACE versus patients treated by TAE. (**A**) Liver transplantation-free survival without adjustment and (**B**) liver transplantation-free survival adjusted using the propensity score in patients treated by TACE versus patients treated by TAE. The results were analyzed using the Kaplan–Meier method and compared using the log rank test. Numbers at risk were reported under the *x*-axis.

**Table 1 cancers-13-00812-t001:** Description of the population.

Patients’ Characteristics	TAE *n* = 86	TACE *n* = 179	Total *n* = 265	*p* Value (TAE Versus TACE)
Center (Grenoble) *	0 (0%)	153 (85.5%)	153 (57.7%)	<0.0001
Gender (Male) *	76 (88.4%)	162 (90.5%)	238 (89.8%)	0.591
Age (years old) ^$^	69 (61–78)	63 (57–71)	65 (58–75)	0.0003
Performance status (0) *	65 (75.6%)	146 (81.6%)	211 (79.6%)	0.968
Etiologies of chronic liver disease *	*n* = 84	*n* = 168	*n* = 252	
High alcohol intake	26 (31.0%)	49 (29.2%)	75 (29.8%)	
NASH	7 (8.3%)	16 (9.5%)	23 (9.1%)	
Chronic hepatitis C	20 (23.8%)	21 (12.5%)	41 (16.3%)	
Chronic hepatitis B	4 (4.8%)	13 (7.7%)	17 (6.7%)	
Co-HBV + HCV	0 (0%)	4 (2.4%)	4 (1.6%)	
Alcohol + HBV or HCV	11 (13.1%)	20 (11.9%)	31 (12.3%)	
Alcohol + metabolic	16 (19.0%)	37 (22.0%)	53 (21%)	
Hemochromatosis	1 (1.2%)	6 (3.6%)	7 (2.7%)	
Other etiology	1 (1.2%)	2 (1.2%)	3 (1.2%)	
Cirrhosis *	82 (95.4%)	159 (88.8%)	241 (90.9%)	0.083
Child Pugh class				0.031
A	77 (89.5%)	145 (81.0%)	222 (83.8%)	
B	9 (10.8%)	34 (19.0%)	43 (16.2%)	
Serum AFP ^$^	7 (4–16)	15 (5–71)	11 (4–48)	0.0045
Sum of size of the 2 main nodules ^$^	62 (42–90)	54 (36–81)	55 (37–83)	0.1131
Number of nodules				<0.0001
Unique nodule *	8 (9.3%)	55 (30.7%)	63 (23.8%)	
2–3 lesions *	17 (19.8%)	62 (34.6%)	79 (29.8%)	
≥4 lesions *	61 (70.9%)	62 (34.6%)	123 (46.4%)	
BCLC stage				<0.0001
A *	9 (10.5%)	64 (35.8%)	73 (27.6%)	
B *	66 (76.7%)	84 (46.9%)	150 (56.6%)	
C *	11 (12.8%)	31 (17.3%)	42 (15.9%)	
Type of treatment at first TAE/TACE				
Bilobar treatment *	74 (86.1%)	130 (72.6%)	204 (77%)	0.050
Lobar treatment *	11 (12.8%)	43 (24%)	54 (20.4%)	
Segmental treatment *	1 (1.2%)	6 (3.4%)	7 (2.6%)	
Number of procedures ^$^	2 (1–2)	2 (1–2)	2 (1–2)	0.7562
Liver transplantation *	7 (8.1%)	44 (24.6%)	51 (19.2%)	0.001
Treatments following TAE/TACE				
Radiofrequency ablation	16 (18.6)	10 (5.6)	26 (9.8)	
Surgical removal	0	1 (0.6)	1 (0.4)	
Systemic therapiesSorafenib	23 (26.7) 22 (25.6)	31 (17.3) 18 (10.1)	44 (16.6) 40 (15.1)	
Clinical trial	1 (1.2)	13 (7.3)	14 (5.3)	
Other treatment (SIRT, alcoholization)	1 (1.2)	11 (6.1)	12 (4.5)	
No additional treatment	19 (22.1)	45 (25.1)	64 (24.2)	

* Number (percentages), ^$^ Median (interquartile range); AFP: alphafetoprotein, BCLC: Barcelona Clinic Liver Cancer, HBV: hepatitis B virus, HCV: hepatitis C virus, NASH: non-alcoholic steatohepatitis, SIRT: Selective Internal Radiotherapy, TAE: trans-arterial embolization, TACE: trans-arterial chemoembolization.

## Data Availability

The data presented in this study are available on request from the corresponding author. The data are not publicly available due to confidential data among data sets.

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
