# Peer review of "Comparison of Trans-Arterial Chemoembolization and Bland Embolization for the Treatment of Hepatocellular Carcinoma: A Propensity Score Analysis"

_cancers, 2021, doi:10.3390/cancers13040812_

Round 1
Reviewer 1 Report
1, in the result section, that authors stated that "the Patients treated by TAE were older (median age: 69 versus 63 ; P=0.0003), had a higher tumor burden (median tumor number: 3 versus 2 ; P< 0.0001), with less BCLC A HCCs (10.5% versus 35.8% ; P< 0.0001), compared to patients treated by TACE. In contrast, patients treated by TAE had a lower Child Pugh score (median: 5 versus 6 ; P< 0.0001) and a lower AFP level (median: 7.0 versus 15.0 ng/mL ; P=0.0045)
compared to patients treated by TACE (Table 1)." There are statistical differences in the patient's age, tumor burden, BCLC, Child Pugh score and AFP levels between the TACE and TAE. The study design can be improved to make the two groups more comparable.
2, 51 patients (19,3%) had liver transplantation after a median follow-up of 21.6 months with more patients treated by TACE receiving transplantation (24.6%) than patients treated by TAE (8.1%, P=0.001). Were there any differences in patient's age, tumor burden, and MELD score? These information would be associated with post-transplant complications and outcome.
3, The aim of this study was to compare radiological response. However, the gold standard should be the pathology, particularly in the 51 patients with transplantation, as the final pathology diagnosis of explant liver may not be consistent with the radiological response. A comparison of radiological response to final pathology diagnosis including numbers of viable tumor nodules vs complete tumor necrosis, TNM stage, vascular invasion, residual tumor grade and cholangiocarcinoma differentiation, etc. could be included.
Author Response
Responses to reviewer 1
- in the result section, that authors stated that "the Patients treated by TAE were older (median age: 69 versus 63 ; P=0.0003), had a higher tumor burden (median tumor number: 3 versus 2 ; P< 0.0001), with less BCLC A HCCs (10.5% versus 35.8% ; P< 0.0001), compared to patients treated by TACE. In contrast, patients treated by TAE had a lower Child Pugh score (median: 5 versus 6 ; P< 0.0001) and a lower AFP level (median: 7.0 versus 15.0 ng/mL ; P=0.0045) compared to patients treated by TACE (Table 1)." There are statistical differences in the patient's age, tumor burden, BCLC, Child Pugh score and AFP levels between the TACE and TAE. The study design can be improved to make the two groups more comparable.
Response: We thank you for this relevant critic which pinpoints a limit of our study. As every patient during this period was included with treatment’s allocation done in a systemic manner due to local practice, indepently to these clinical features, this limit is inherent to the retrospective character of this work. Besides, the use of a propensity score weighted on the age, Child-Pugh score, AFP level, the sum of the 2 main liver nodules and the number of tumors was performed to decrease the bias due to these different in the statistical analyses.
- 51 patients (19,3%) had liver transplantation after a median follow-up of 21.6 months with more patients treated by TACE receiving transplantation (24.6%) than patients treated by TAE (8.1%, P=0.001). Were there any differences in patient's age, tumor burden, and MELD score? These information would be associated with post-transplant complications and outcome.
Response: Dear reviewer, we thank you for these very interesting remark . We believe that these analyses may be integrated in a new different work about the impact of pre transplantation procedures on post LT outcomes.
- The aim of this study was to compare radiological response. However, the gold standard should be the pathology, particularly in the 51 patients with transplantation, as the final pathology diagnosis of explant liver may not be consistent with the radiological response. A comparison of radiological response to final pathology diagnosis including numbers of viable tumor nodules vs complete tumor necrosis, TNM stage, vascular invasion, residual tumor grade and cholangiocarcinoma differentiation, etc. could be included.
Response: Dear reviewer, to our knowledge, there is no validated tumor regression score in HCC than can be used to estimated tumor response by histology, contrary to colorectal cancer for example. Thus, comparison of tumor size by histology, to baseline tumor size by radiology is not a standard. In this work, we used mRECIST criteria as recommended by international guidelines. Besides, this analysis is performed at 3 months to study TACE and TAE perfomances, when patients waiting for transplantation are transplanted after a median delay 18 to 24 months in France. In the latter situation, tumor tissue residue on liver explant after such a long period, is not a good estimation of TACE/TAE efficacy, as patients may have several other treatments before liver transplantation, and time of waiting, as well as the degree of liver insufficiency are essential parameters in neo adjuvant treatments before liver transplantation that may consitute bias. To finish, liver transplanted patients represent a small part of our cohort, especially in the TAE group, suggesting that this parameter cannot be used to compare both techniques due to a lack of statistical power.
Reviewer 2 Report
The authors present a retrospective analysis with the aim to compare TACE to TAE in patients with intermediate-stage HCC by using a propensity score methodology.
Overall, the paper is well written and the data are well presented. The main limitation is the relatively low number of pts analyzed and, therefore, the lack of statistical power to draw strong conclusions despite the propensity score weighting.
Another important limitation is the long study period (from 2007 to 20014) coupled with a relatively short median follow up period (9 months), furthermore, the lack of selection criteria between the 2 procedures (the center policy was the only criteria used) is a possible source of bias.
I suggest the following revisions:
- In the discussion sections, authors should comment the long study duration and should also underline the limitation of a short follow up period
- In the results section, authors should add the median number of procedures per treatment arm
- An important piece of information in this kind of trial is also the time to liver decompensation, it could be added?
- If available, it should be useful, to better interpret the results, to report the treatments after progression
Author Response
Responses to reviewer 2
I suggest the following revisions:
- In the discussion sections, authors should comment the long study duration and should also underline the limitation of a short follow up period
Response: Dear reviewer, we agree that a long period of inclusion usually constitutes a bias due to practice changes with technical advances or improvement of the management of post-operative adverse events. Nonetheless, in this work periods of inclusion are respectively 3 and 6 years in Grenoble and Jean Verdier centers, that we considere as reasonable to study techniques in which no significant advances were obtained during this period.
The follow-up time has been corrected and is 21.7 months.
- In the results section, authors should add the median number of procedures per treatment arm
Response: Dear reviewer, this information is already mentioned in the text in the section « 3.1. Clinical features of the population ». We had this information in table 1.
- An important piece of information in this kind of trial is also the time to liver decompensation, it could be added?
Response: Dear reviewer, due to the retrospective character of this work, we are not able to perform this long term safety analysis due to missing data.
- If available, it should be useful, to better interpret the results, to report the treatments after progression
Response: Dear Reviewer, these informations have been added to the manuscript in table 1.
Reviewer 3 Report
I found this manuscript has broad clinical research data. This might be useful for further HCC treatment regimen. All the figures must be revised with higher contrast to make more clear. This manuscript must be thoroughly check for typo-errors.
Author Response
Responses to reviewer 3
I found this manuscript has broad clinical research data. This might be useful for further HCC treatment regimen. All the figures must be revised with higher contrast to make more clear. This manuscript must be thoroughly check for typo-errors.
Response: Dear reviewers, these issues have been adressed.
Reviewer 4 Report
This nicely presented retrospective analysis addresses an interesting and not yet solved question: What is the best liver-directed treatment for BCLC-B HCC patients? According their results, non significant advantages are obtained by adding chemotherapy to the embolization procedures, exception made of complete response rate.
These are some suggestions to improve the paper:
- Subsequent treatments after embolization should be described. If there were no additional treatments, it should be specified.
- Histological response evaluation in liver specimens would permit to asses the accuracy of radiological complete response evaluation.
Minor corrections:
Line 44: blan – bland
Line 11: drug administration, was immediately – drug administration was inmmediatly
Line 122: 2.4. Data collection: - 2.4. Data collection
Line 168: (IQR = 168 1;2) – Explain abbreviature
Line 193: child-pugh score – Child-Pugh score
Author Response
Responses to reviewer 4
This nicely presented retrospective analysis addresses an interesting and not yet solved question: What is the best liver-directed treatment for BCLC-B HCC patients? According their results, non significant advantages are obtained by adding chemotherapy to the embolization procedures, exception made of complete response rate.
These are some suggestions to improve the paper:
- Subsequent treatments after embolization should be described. If there were no additional treatments, it should be specified.
Response: Dear Reviewer, these informations have been added to the manuscript in table 1.
- Histological response evaluation in liver specimens would permit to asses the accuracy of radiological complete response evaluation.
Response: Dear reviewer, to our knowledge, there is no validated tumor regression score in HCC than can be used to estimated tumor response by histology, contrary to colorectal cancer for example. Thus, comparison of tumor size by histology, to baseline tumor size by radiology is not a standard. In this work, we used mRECIST criteria as recommended by international guidelines. Besides, this analysis is performed at 3 months to study TACE and TAE perfomances, when patients waiting for transplantation are transplanted after a median delay 18 to 24 months in France. In the latter situation, tumor tissue residue on liver explant after such a long period, is not a good estimation of TACE/TAE efficacy, as patients may have several other treatments before liver transplantation, and time of waiting, as well as the degree of liver insufficiency are essential parameters in neo adjuvant treatments before liver transplantation that may consitute bias. To finish, liver transplanted patients represent a small part of our cohort, especially in the TAE group, suggesting that this parameter cannot be used to compare both techniques due to a lack of statistical power.
- Minor corrections:
Line 44: blan – bland
Line 11: drug administration, was immediately – drug administration was inmmediatly
Line 122: 2.4. Data collection: - 2.4. Data collection
Line 168: (IQR = 168 1;2) – Explain abbreviature
Line 193: child-pugh score – Child-Pugh score
Response: Dear reviewers, these issues have been adressed.
Round 2
Reviewer 1 Report
The normal liver receives a dual blood supply from the hepatic artery (25%) and the portal vein (75%). As HCC grows, it increasingly depends on the hepatic artery for blood supply and once a tumor nodule reaches a diameter of 2 cm or more, most of the blood supply derives from the hepatic artery. This unique property of HCC provides the rationale for the use of transarterial therapies. TACE and TAE consist of the selective angiographic occlusion of the tumor arterial blood supply with a variety of embolizing agents, with or without the precedence of local chemotherapy infusion. The occlusion by embolic particles results in tumour hypoxia and necrosis, while the addition of local chemotherapy could have an additive anti-tumour effect. The efficacy of TA(C)E was established by at least 9 randomized controlled trials (RCTs). The meta-analysis was heavily criticized for the exclusion of positive RCTs due to risk of bias and inappropriate inclusion of trials. The inconsistency in the results of randomized trials reflects the fact the TA(C)E was likely due to bias of patient selection and the procedure itself. This manuscript (1054310) is not randomized conrolled study. The patient selection is inappropriate.
Author Response
Reviewer 1
The normal liver receives a dual blood supply from the hepatic artery (25%) and the portal vein (75%). As HCC grows, it increasingly depends on the hepatic artery for blood supply and once a tumor nodule reaches a diameter of 2 cm or more, most of the blood supply derives from the hepatic artery. This unique property of HCC provides the rationale for the use of transarterial therapies. TACE and TAE consist of the selective angiographic occlusion of the tumor arterial blood supply with a variety of embolizing agents, with or without the precedence of local chemotherapy infusion. The occlusion by embolic particles results in tumour hypoxia and necrosis, while the addition of local chemotherapy could have an additive anti-tumour effect. The efficacy of TA(C)E was established by at least 9 randomized controlled trials (RCTs). The meta-analysis was heavily criticized for the exclusion of positive RCTs due to risk of bias and inappropriate inclusion of trials. The inconsistency in the results of randomized trials reflects the fact the TA(C)E was likely due to bias of patient selection and the procedure itself. This manuscript (1054310) is not randomized conrolled study. The patient selection is inappropriate.
Response: Dear reviewer, we thank you for this comment and we totally agree on the fact that randomized clinical trial would be the best option to answer to the question « is TACE superior to TAE ? », even thought previous studies let this question unresolved due to suboptimal methodological plans. Nonetheless, this bicentric retrospective study included every patients undergoing TACE or TAE, giving a homogenous population and used a propensity score matching to decrease biases influence on results and increase comparability.
Reviewer 2 Report
I think that the manuscript has been adequately improved and now warrants publication in Cancers
Author Response
.